# The Diverse Functions of Mutant 53, Its Family Members and Isoforms in Cancer

**DOI:** 10.3390/ijms20246188

**Published:** 2019-12-07

**Authors:** Callum Hall, Patricia A.J. Muller

**Affiliations:** CRUK Manchester Institute, Manchester SK10 4TG, UK; callum.hall@cruk.manchester.ac.uk

**Keywords:** mutant p53, gain-of-function, p53 family

## Abstract

The p53 family of proteins has grown substantially over the last 40 years. It started with p53, then p63, p73, isoforms and mutants of these proteins. The function of p53 as a tumour suppressor has been thoroughly investigated, but the functions of all isoforms and mutants and the interplay between them are still poorly understood. Mutant p53 proteins lose p53 function, display dominant-negative (DN) activity and display gain-of-function (GOF) to varying degrees. GOF was originally attributed to mutant p53′s inhibitory function over the p53 family members p63 and p73. It has become apparent that this is not the only way in which mutant p53 operates as a large number of transcription factors that are not related to p53 are activated on mutant p53 binding. This raises the question to what extent mutant p53 binding to p63 and p73 plays a role in mutant p53 GOF. In this review, we discuss the literature around the interaction between mutant p53 and family members, including other binding partners, the functional consequences and potential therapeutics.

## 1. Introduction

In 1979 the research into p53 started off with the finding of a 53kDa protein using immunoprecipitation with the SV40 T antigen in transformed cells [1,2,3]. Forty years on, two other family members, p63 and p73, a large variety of different isoforms with altered N- and C-termini and last, but not least, thousands of different p53 mutant proteins have been identified. p53 has been found to be activated by a large variety of stresses and has been shown to regulate myriad genes, hence working as a tumour suppressor. It is thought that p53′s response allows for survival and adaptation to mild stresses, but will initiate cell death if the stress is too much [4]. Stresses include, but are not limited to, DNA damage, metabolic stress, or ribosomal stress (recently reviewed in [5]).

The importance of p53 in tumour suppression is apparent when looking at the frequency of p53 mutations in all cancers, the early onset and frequency of cancers in Li-Fraumeni patients and the many mouse models that display tumourigenesis when p53 is lost or mutated. Using data from the TCGA (The Cancer Genome Atlas), it has become apparent that more than 50% of all cancers carry p53 mutations, but in some cancers, such as small cell lung cancer and ovarian cancer, this percentage can go up to levels higher than 90% [6,7]. Mutations in *TP53* can cause loss of p53 expression or expression of mutated proteins. The mutant proteins are almost always caused by single nucleotide changes in the DNA, resulting in mutant proteins that differ in only one amino acid from the wildtype molecule. Unlike many other tumour suppressors, mutations are found throughout the amino acid sequence of p53, although more frequently in the DNA binding domain, giving rise to thousands of different mutant proteins [8]. The most common mutant p53 proteins (the hotspots) can be regarded as inactive cousins of the wildtype protein, but screens in yeast have indicated that not all mutants are completely impaired in wildtype function. A large variety in the extent to which each mutant p53 protein can still operate as a wildtype molecule exists [9,10].

To function as a transcription factor, p53 forms dimers and tetramers that interact with the DNA. The observation that mutant p53 is often impaired in wildtype function led to the observation that it can exert a dominant-negative role over any remaining wildtype p53 protein through direct binding [11,12]. In addition to dominant-negative activity, it is now apparent that many mutant proteins can also exert a gain-of-function (GOF) in promoting tumourigenesis, inducing metastasis, engulf neighbouring cells and avert cell death induced by chemotherapeutics or other stressors [13,14,15,16]. Halevy et al. first described the three major differences between mutant p53 and wildtype p53. Mutant p53 proteins lose tumour suppressor function, have an increased transforming potential and show increased stability. These functions are independent of each other and are not exhibited by each mutant to the same extent [17]. GOF was further demonstrated in mouse models in which expression of mutant p53 caused a more aggressive tumour profile than the loss of p53 expression [18,19]. Increased stability of the mutant might be a prerequisite for gain-of-function as normal levels of mutant p53 are observed in most tissues of Li-Fraumeni patients and mutant p53 mouse models, whereas most tumours overexpress mutant p53 [20,21]. This is substantiated by the finding that stresses that stabilise the wildtype p53 protein also promote mutant p53 stabilisation and therefore, potentiate tumourigenesis in mouse models [22]. The mechanisms underlying GOF are numerous and often involve a capacity of mutant p53 to bind proteins that wildtype p53 does not interact with. This was first demonstrated for the heat shock protein complex [23], but now includes numerous other proteins, including the p53 family members p63 and p73. In this review, we will focus on how mutant p53 inhibits p63 and p73 and to what extent this inhibition contributes to mutant p53 gain-of-function in vivo.

## 2. The p53 Family of Proteins

Shortly after *TP53*, the family members *TP63* and *TP73* were cloned and identified as homologues of p53 that share a similar DNA binding domain but differ in N- and C-termini. All family members can be transcribed from different promotors resulting in variants with long N-termini that contain transcription activation domains (TA) or shortened N-termini (ΔN) (Figure 1a). TA isoforms can bind to canonical p53 target sequences and can on overexpression induce the expression of p53 target genes [24,25,26]. Despite lacking a transactivation domain, ΔN forms are impaired in promoting the expression of TA target genes, but they are not transcriptionally inactive. They have been shown to regulate the expression of their own set of genes and promote tumourigenesis [27,28,29,30]. C-terminal variations occur due to alternative splicing giving rise to over 20 different isoforms. Full-length versions of p63 and p73 (p63α and p73α) contain a C-terminal oligomerisation (OD) domain, a sterile alpha motif (SAM) domain and a transcription inhibitory domain (TID) (Figure 1a). Splice variants of p63 and p73 (β, χ, δ, ε and others), as well as all p53 isoforms, are lacking the SAM domain and the TID (Figure 1a,b).

p63 and p73 are thought to be expressed in a more tissue-specific manner than p53, playing important roles in the skin epithelium and in the central nervous system. This is reflected in the phenotypes of *Tp63* KO mice and *Tp73* KO mice that show lethality during late gestation or early after birth due to skin and craniofacial epithelial defects [31,32] or neurological defects, respectively [33] Although most *Tp53* knockout embryos survive until early onset of tumourigenesis [34], subtle defects in embryonic development can be noted [35], suggesting distinct functions in embryonal development for each family member.

In comparison to *TP53*, mutations in *TP63* and *TP73* are rarely found in cancers. This raises the question to what extent p63 and p73 contribute to tumourigenesis. Although mice that are knockout for all p63 isoforms or all p73 isoforms show early lethality, loss of TAp63 or TAp73 expression specifically promoted tumourigenesis or metastasis [36,37,38]. Furthermore, loss of ΔN isoforms of p63 and p73 fully recapitulated the lethality phenotype of full isoform knockouts. These data indicate the importance of ΔN isoforms in development, but do not exclude the possibility that these isoforms play a role in tumourigenesis. Indeed, ΔN isoforms have been found to be important in tumour formation and can be found overexpressed in various types of cancers [39,40]. More importantly, Venkatanarayan et al. demonstrated that conditional loss of ΔNp63 or ΔNp73 increased the lifespan of p53 knockout mice that would otherwise succumb to tumour formation [30]. Loss of ΔNp63 or ΔNp73 in these animals led to the upregulation of the expression of TAp63 and TAp73, and so prevented tumour formation [30].

Under normal conditions, ΔN isoforms dominant-negatively interact with TA forms of their family counterparts, and isoforms can transcriptionally regulate each other [30,41,42]. As an example, full-length p53 can regulate an internal promoter for two ΔNp53 Δ133isoforms (Δ133p53 and Δ160p53) expression, but Δ133p53 was also shown to regulate the expression of ΔNp63 variants and ΔNp73, making it very complicated to attribute functions of a specific isoform solely to that isoform [41]. In general, many of the ΔN isoforms seem to exert a pro-tumourigenic role, while TA variants seem to exert tumour suppression through activation of canonical p53 target genes. However, the role of each isoform and especially C-terminal variants, as well as the interplay between all isoforms, warrants further research.

## 3. The Interactions between Mutant p53 and p63 or p73

In addition to inhibiting wildtype p53, many mutant p53 proteins interact with most isoforms of p63 and p73 (Figure 1) [43,44,45]. In this respect, it is important to note that the interaction between mutant p53 and family members is more pronounced for those p53 mutants that are structurally unfolded [43,44,45]. Although most research has, therefore, focused on the quintessentially unfolded R175H mutant p53, even mutants that are considered to be normally folded, such as the R273H, can interact with p63 or p73, albeit to a lesser extent [46,47]. It is also interesting to note that a decreased binding of a particular mutant p53 to p73 might not always predict how well it binds to p63. Strano et al. identified that the 281G mutant p53 can interact with p73, but cannot interact with p63 [44,45], which is consistent with the observation that loss of amino acids 251–312 in R175H mutant p53 impairs p63 binding, but not p73 binding [46].

Interactions between mutant p53 and p63 or p73 can be observed in vitro and are strong, indicating that mutant p53-p63 or mutant p53-p73 bind directly with each other [44,45,48]. For the p63-mutant p53 R175H interaction, the DBD of either molecule was able to interact with the full-length version of the other molecule, suggesting that this binding does not require the OD in either protein [44]. However, loss of the C-terminus of p63 (β or χ isoforms) did markedly reduce the interaction [46,49]. For p73, a p73 molecule containing the DBD + OD was sufficient to bind to mutant p53 R175H, and the DBD of the R175H mutant p53 molecule was sufficient to interact with p73α and p73χ [45] (Figure 1b,c). However, p73β or a variant lacking the transcription inhibitory domain (TID) was impaired in binding to mutant p53 R175H [45,46]. These data together point to multiple ways in which mutant p53 R175H can interact with any of the TA isoforms of the p53 family proteins, but a note of caution has to be made on the interpretation of all interaction mapping studies. The use of deletion mutants can affect the structure of the protein and result in changes in binding that are not necessarily associated with that particular region. An example of how structure can influence binding is shown by Kehrloesser et al., who discovered that TAp63α forms a closed dimer in which an N-terminal region interacts with a C-terminal region, leading to a decrease in the ability of mutant p53 to bind this isoform [49]. Loss of the N-terminus of p63α restores binding to mutant p53 R175H by preventing the formation of the closed dimer of p63.

Much work on mutant p53 interactions has so far focused on delineating how mutant p53 proteins inhibit TA variants. All TA isoforms of the p53 family contain an oligomerisation domain (OD) that is used for homo-dimerisation and tetramerisation. Remarkably, heterodimerisation between TA isoforms of the different family members is seldom observed under physiological circumstances. Mutant p53 R175H can dominant negatively inhibit wildtype p53 by binding through the OD [50]. However, in vitro, the interaction between mutant p53 R175H and wildtype dissociates less easily than the interaction between two wildtype molecules [48]. These data, together with the fact that other domains have been identified that impair mutant p53 R175H from interacting with wildtype p53 [51], suggest that mutant p53 R175H represses wildtype p53 by binding to this molecule in at least two different ways.

Some of the mouse models that were generated of N-terminally truncated p53 display a similar phenotype to mutant p53 (R172H or R270H) in promoting invasion and metastasis [52]. As Δ133p53 can interact and potentiate ΔNp63 function, perhaps the interaction between mutant p53 and ΔNp63 and ΔNp73 variants as well as expression of all p53 family isoforms should be more thoroughly investigated in these models [53]. Notably, mutant p53 R175H can readily interact with both p63 and p73 ΔN isoforms [46,49]. Instead of inhibiting these isoforms, it has been observed that mutant p53 R175H can form a complex with ΔNp73 to promote ΔNp63 expression [54]. As mentioned before, N-terminal variants are readily detected in human cancers, and some of these variants correlate with protein expression of mutant p53. As we are only beginning to understand the function of mutation of p53 in the full-length variant of the p53 molecule (TAp53α), at the minute, it is only guesswork on how intrinsic mutations affect ΔNp53 function.

## 4. Functional Consequences and Binding Partners of Mutant p53-TAp63 and Mutant p53-TAp73 Complexes

To investigate how mutant p53 inhibits p63 and p73 function, several laboratories have studied potential binding partners of the mutant p53-family member complexes (Figure 2). These studies were almost exclusively done with the TA variants of p63 and p73 and with the R175H mutant p53, although some other mutants were investigated that did show similar binding propensities.

Some of these binding partners were found to act in a similar manner in both the mutant p53-p63 and mutant p53-p73 complexes. The topoisomerase TopBP1, for instance, promotes the formation of both complexes by binding p53 and increasing the affinity for mutant p53 to bind to p63 or p73. This leads to loss of p63 or p73 mediated transcription and an increase in invasion and metastasis [55]. Conversely, ANKRD11 dissociates both complexes, restoring p63 and p73 mediated transcription and a subsequent lower incidence of invasion and metastasis [56].

Interestingly, the binding partners of the complexes do not always act on p63 or p73 in the same way. MDM2 and HSP70 differentially regulate the two complexes by promoting the dissociation of the mutant p53-p63 complex but enabling the formation of the mutant p53-p73 complex. This results in a loss of p73-mediated transcription and a gain in p63-mediated transcription [46,57]. Interestingly, a closer examination of the mutant p53-p73 complex further revealed that other HSP family members can also interact with the complex and that MDM2 can replace the HSP proteins to form a more stable complex and potentiate chemoresistance [58].

Most binding partners have been shown to regulate only p63 or p73. For instance, SMAD and Pin1 promoted the mutant p53-p63 complex formation, inhibiting p63 function [36,59]. The transcription factor ATF3 has been shown to prevent the formation of the mutant p53-p63 complex with a resultant suppression of mutant p53-mediated invasion and metastasis [60]. For the mutant p53-p73 complex, JNK was found to promote the formation of the mutant p53-p73 complex leading to the loss of p73 mediated transcription [61]. For all these binding partners, it will be interesting to see to what extent they regulate the transcriptional function of just one of the p53 family members or whether they can also regulate the other family member.

These findings show that the interacting proteins of mutant p53-family member complexes in cancer can easily modulate some aspects of mutant p53 GOF. This regulation, if appropriately targeted, could be used to reduce the GOF of mutant p53 that is related to p63 and p73 function. In particular, looking at proteins that affect both complexes in the same manner could be interesting. Inhibition of TopBP1 or restoration of ANKRD11 expression would both lead to wild type p63 and p73 signalling, which would suppress metastasis and induce apoptosis.

## 5. Mutant p53 Aggregation and Other Binding Partners

Over the last few years, the interaction between mutant p53 and family members has gotten significant attention from researchers interested in prion-like aggregation. In vitro, the DBD domain was shown to form aggregates, and the OD and mutant forms of p53 were prone to form amyloid [62,63]. Xu et al. proposed a region around amino acid 254 to be required for aggregation of p53 in vivo, but in vitro aggregation was shown to depend on multiple regions and could still occur when this region was mutated [51,64]. In addition, p63 and p73 were shown to have intrinsic aggregation properties [49,65]. Given that mutant p53 is inhibiting the function of TAp63 and TAp73, it has been postulated that through aggregation, mutant p53 might inactivate its family members, preventing them from interacting with their target gene promoters [51,66]. Aggregation was identified as punctate staining for p53 in the cytoplasm or nucleus [67]. A punctate p53 staining has been identified in cancers with mutant or wildtype p53 expression and has been correlated to worse outcomes [67]. With the notion that p53 can form aggregates comes the hypothesis that p53 might be a prion protein. In this hypothesis, p53 protein aggregates would grow until causing toxicity and cell death, leading to the release of the aggregates into the cell surroundings and spread to other cells. Forget et al. showed that aggregated N-terminally truncated p53 can enter cells and promote aggregation of endogenous p53 [68]. Whether or not p53 is secreted by cells as aggregates or released into the microenvironment upon necrosis and so spreads remains to be elucidated.

Most likely, aggregation cannot be fully responsible for all gain-of-function that we see on mutant p53 expression. By aggregating, the mutant p53 complex with all of its binding partners is rendered inactive. Although mutant p53 inhibits TAp63 in activating canonical p53 target genes, some reports describe that through mutant p53, TAp63 binds to a unique set of promotors to promote gene expression [69,70]. In addition, mutant p53 interacts with a wide variety of other transcription factors and proteins to promote invasion, acquire chemoresistance or to enhance tumourigenesis [13,71]. These include, but are not limited to, HIF-1, ETS members and SREBPs [72,73,74]. Most strikingly, an interaction of mutant p53 with many of these transcription factors potentiates the function of these transcription factors, illustrating that mutant p53 or the mutant p53 complex is not always inactivated through aggregation.

## 6. The Role of the p53 Family in Tumour Formation in Mouse Models

As mutant p53 has been shown to exert gain-of-function in many ways, it will be important to determine to what extent the interaction between mutant p53 and p53 family members, as well as with other molecules, plays a role in the different aspects of tumourigenesis in vivo. Most mutant p53 mouse models have investigated hotspot mutations that are generally considered fully impaired in wildtype function, stabilised in tumours and capable of GOF [18,19]. These models generally show more metastasis and a broader aggressive tumour burden compared to p53 knockout mice, but differences between the potency of different mutant p53′s have been noted [75]. Loss of TAp63 and loss of TAp73 expression have been shown to cause a similar increase in tumourigenesis, as seen in mutant p53 models [37,38]. Given that mutant p53 can inhibit these family members as well as enhancing the function of other transcription factors, this raises the question to what extent mutant p53 exerts its GOF through inhibition of its family members. In the KPC mouse model in which mutant p53 (R172H) is specifically expressed in the pancreas in combination with an activating *kRAS* mutation, tumours were found to frequently metastasise to the liver and adjacent tissues. Loss of p53 expression in combination with loss of TAp63 expression also led to metastasis, but with a lower prevalence, suggesting that loss of TAp63 does not exclusively contribute to mutant p53 GOF in metastasis in this model [76]. In addition, *TAp63* and *TAp73* knockout animals have been reported to display additional phenotypic abnormalities in ageing, metabolism or spermatogenesis, respectively, that have so far not been reported to occur in any of the mutant p53 mouse models [77,78,79]. These data suggest that mutant p53 does not exclusively operate through inhibition of TAp63 and TAp73 and that perhaps only part of the functions of TAp63 and TAp73 is inhibited by mutant p53.

To fully elucidate how the interactions between mutant p53 and family members work together in promoting tumourigenesis and metastasis, it will be important to combine loss of TAp63 expression with mutant p53 in the same model system and determine whether the tumour burden is exacerbated. For this, it will also be important to fully characterise the expression of isoforms of the p53 family. Loss of ΔN isoforms of p63 and p73 in *Tp53* KO mice was shown to inhibit tumourigenesis in p53 knockout animals by upregulating TA isoforms [30]. It will be interesting to see if this can also happen in mutant p53 mice or whether mutant p53 prevents the TA forms from working in this model.

## 7. Therapeutic Options to Restore TAp63 and TAp73 Function in Mutant p53 Cells

Although p53 has been studied for 40 years and is the most mutated gene in all cancers, hardly any therapies or clinical decisions are based on p53 status. This is not due to a lack of effort in trying to design clever ways to combat mutant p53 GOF or to increase wildtype function. Stabilisation of wildtype p53 has been thoroughly studied, with nutlin-mediated inhibition of MDM2 (the E3 ubiquitin ligase involved in p53 degradation) being one of the most famous protein–protein interference drugs [80]. Therapeutic strategies in mutant p53 cells include attempts to decrease mutant p53 expression, prevent mutant p53 from binding to target proteins, target downstream signalling molecules, restore the normal folding of the mutant protein or viral delivery of wildtype p53 or perhaps family members (summarised and discussed in [81,82,83]). Here, we will only highlight a few selected therapies relevant for the mutant p53-TAp63 and mutant p53-TAp73 interaction, specifically, summarised in Table 1.

Remarkably, there are quite a few drugs that prevent mutant p53 from binding to TAp73, but no therapeutics that directly and specifically interfere in the mutant p53-TAp63 interaction (Table 1). This raises questions as to whether the research is somehow biased towards finding p73 targets, whether mutant p53-p73 targets would also abolish mutant p53-p63 interaction and if the mutant p53-p63 interaction is less suitable to be targeted with small molecules. Most of the mutant p53-p73 interfering drugs were found through drug library screens with the aim to induce apoptosis in mutant p53 cells and not in p53 null or p53 wildtype cells. RETRA, Peptide Aptamers, LEM2 and Prodigiosin were reported to increase p73 function specifically and work independently of p63 function [84,85,88,89]. Although overexpression of TAp63 has been found to promote the expression of some of the p53 target genes, TAp63 might be less potent in inducing p53 target genes, such as *BAX* or *GADD45* [98]. These data could, therefore, indicate that the mutant p53-TAp73 interaction is specifically important in the chemoresistance and prevention of cell death GOF phenotype, whereas the mutant p53-TAp63 interaction might be more important in invasion and metastasis. As it is harder to set up large drug discovery screens for invasion and metastasis, research might, therefore, be biased towards finding apoptosis enhancers and thus favour mutant p53-TAp73 interfering drugs. However, although p63-dependent effects of the inhibitors were not seen in the screens, it is unknown if RETRA, LEM2 or Prodigiosin actually abolish the interaction between mutant p53 and TAp63 [84,88,89]. Perhaps p63 is expressed under the detection limit, or perhaps ΔN isoforms are expressed, and therefore, no TAp63-dependent apoptosis could be seen.

TAp73′s role is certainly not limited to apoptosis as it has been shown to play a role in invasion in pancreatic cancer cells via a protein–protein interaction with NF-Y [91]. It would, therefore, be interesting to systematically determine how each of the compounds that interfere in the mutant p53-p73 interaction affect levels and function of all p53 family isoforms, the interaction of mutant p53 with other transcription factors and the extent of GOF of mutant p53 (e.g., chemoresistance, invasion and metabolism). Ideally, mutant p53-p63 dependent inhibitors should be generated, possibly using domain 251–316 in mutant p53, as loss of that domain specifically prevents TAp63 binding to mutant p53, but does not affect its binding to TAp73 (Figure 1b) [46].

Several compounds have been found that restore the folding of mutant p53 to promote wildtype function, which are listed in Table 2.

Many of these compounds will, therefore, also abolish mutant p53 from binding to the family members, but this is not always specifically investigated. The most well-known example of a drug that restores p53 function is Prima-1, which completed a Phase 1 clinical trial [118,119]. Apart from refolding mutant p53 and restoring p53 function, Prima-1 has been shown to restore TAp73 function [120] and prevent mutant p53 aggregation [120]. As aggregation could mechanistically underlie the inhibition of mutant p53 on p53 and family members, several researchers have looked for and identified inhibitors of mutant p53 aggregation. The aggregation inhibitor ReAcp53 has been shown to inhibit tumour growth of xenografted pancreatic tumour cells [121], but it remains to be investigated if this drug and other aggregation inhibitors affect p63 and p73 function.

## 8. Conclusion and Discussion

An important consideration for this review is that only a handful of p53 mutants, predominantly the hotspot mutations, have been thoroughly investigated. Due to large cancer genome sequencing efforts, we now know that thousands of different mutant p53 proteins exist in cancers. Traditional classification into the DNA contact mutants and the structurally unfolded mutants has not been able to attribute specific functions or cancer characteristics to either one of these classes of mutants. Although unfolded p53 mutants were more able to interact with p63 and p73 than DNA binding mutants, both types of mutants have been shown to inhibit p63 and p73 function. In addition, RETRA and peptide aptamers were able to increase TAp73 function in cells expressing the folded mutant p53 R273H [84,85]. Importantly, the folding state of a protein is relative, and even the wildtype protein has been shown to be able to unfold in certain conditions. An example of this is hypoxia, which was shown to unfold the wildtype p53 protein [122]. It is, therefore, likely that tumours that we presume to express a normally folded protein based on DNA sequencing, such as wildtype or mutant p53 273H, express unfolded proteins based on how hypoxic that tumour is. Although we can detect unfolding in vitro and in cell lines, detecting the folding state in a tumour is much harder. Many of the standard tissue preservation techniques, including formalin and freezing, can unfold p53 making it difficult to investigate folding state and to use it clinically.

Instead of using the folding state of p53, perhaps we can classify mutants in a different way. Although the traditional six hotspots can be detected in almost all cancers with p53 mutations, certain cancers seem to have a predisposition for certain mutants. An example of this is hepatocellular carcinoma (HCC), in which there appears to be a selection for expression of one hotspot, mutant p53 R249S [123]. Remarkably, homozygous mutant p53 R246S (the mouse equivalent of R249S) expressing mice did not display tumourigenesis, as seen in the R172H (the equivalent of R175H) mice. These data suggest additional factors must be involved in the GOF and selection of this mutant. Indeed, it was found that mutant p53 R249S is highly associated with dietary aflatoxin B exposure and HBV infection, which possibly allow for posttranslational modifications of this specific mutant. Similar to HCC, different distributions of p53 mutations can be detected in different types of cancers. It will be interesting to investigate whether other mutations can be selected for dependent on changes in the cancer microenvironment. Libraries, such as created by Kotler et al., could play an important role in discovering which mutants are selected for under which circumstances and could give us valuable information [124]. To use this information in a clinically relevant manner, we will also need to better understand which p53 mutants regulate which effector proteins and what the functional consequences of those mutants are in a cancer type-specific manner.

## Figures and Tables

**Figure 1 ijms-20-06188-f001:**
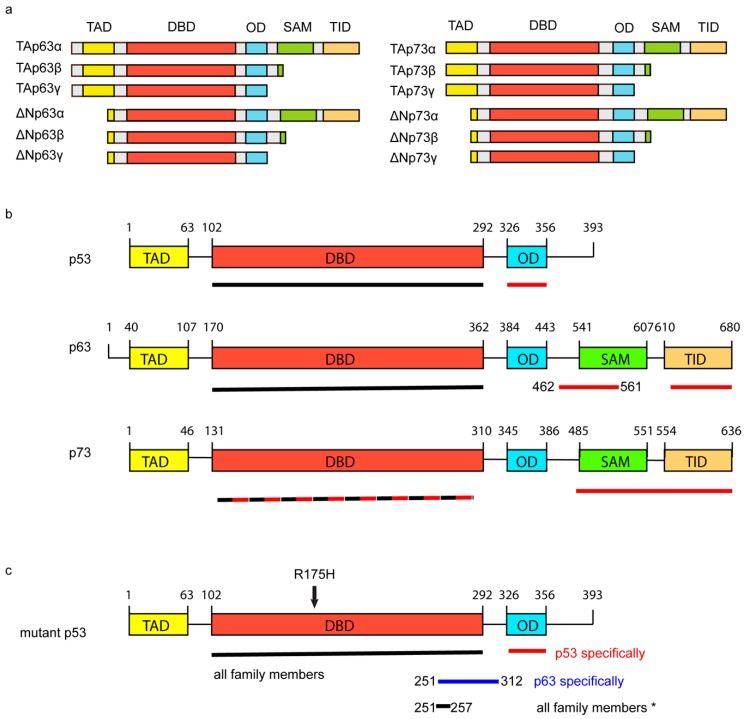
Schematical overview of mutant p53, p53, p63 and p73. (**a**) Overview of the conserved domains in p63 and p73 isoforms TA and ΔN α ,β and χ. (**b**) Conserved domains in p53, p63 and p73 are shown for full-length versions of each protein. Underlined are the areas to which mutant p53 175H has been shown to interact. Black lines indicate studies in which the indicated domain was found on its own to interact with mutant p53. Red lines indicate loss of binding of the indicated p53 family member to mutant p53 R175H when deletion mutants for that specific region were used. TAD = transcription activation domain, DBD = DNA binding domain, OD = oligomerisation domain, SAM= sterile alpha motif, TID = transcription inhibition domain. (**c**) Schematic of conserved domains in full-length mutant p53 (R175H). Underlined are the areas to which full-length versions of p53 family members have been shown to interact. Just the DBD has been shown to interact with all family members. Loss of the OD domain or loss of amino acids 251–312 specifically reduced the interaction of mutant p53 R175H with p53 or p63, respectively. Oligomerisation with all family members was shown to be inhibited by the deletion of the 251–257 region in mutant p53 (R175H). The R175H mutation is indicated in the molecule.

**Figure 2 ijms-20-06188-f002:**
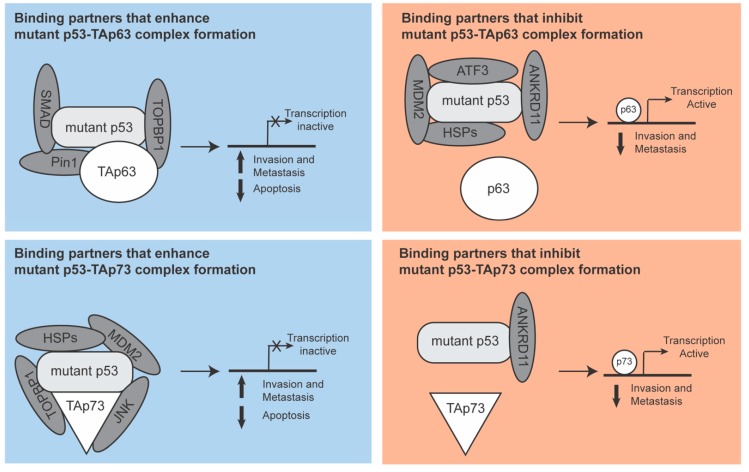
Binding partners of mutant p53-TAp63 and mutant p53-TAp73 complexes. In this figure, binding partners of the mutant p53-p53 family member complexes are depicted. The functional consequence of the binding of each partner on p63 or p73 transcriptional function is shown. Binding partners of the mutant p53-p53 family member complex either promote or inhibit the function of p63 or p73. Some binding partners promote the complex formation between mutant p53 and the p53 family members, sequestering those family members and preventing them from mediating their transcription function. Other proteins prevent the binding of mutant p53 to family members, leading to the release of p63 or p73 and subsequent initiation of transcription, resulting in apoptosis and inhibition of invasion and metastasis. Of note, in most cases, it is not known to which molecule the interaction partners bind. This means that the exact position in which the interaction partner is depicted in this model might not be a precise reflection of the complex in reality. In addition, many different binding partners are shown to interact with the complexes, but it is currently unknown if this co-occurs or whether the binding is mutually exclusive.

**Table 1 ijms-20-06188-t001:** Inhibitors affecting the mutant p53-TAp63 or mutant p53-TAp73 complex and function of p63 or p73.

Drug	Action	Consequence	Tissue/Cells	Ref.
RETRA	Inhibit mutant p53-p73 interaction	Restore TAp73 function	Various cancer cells	[84]
Peptide aptamers	Inhibit mutant p53-p73 interaction	Restore TAp73 function	Tumour cells	[85]
SIMPs *	Inhibit mutant p53-p73 interaction	Restore TAp73 function	Various cancer cells	[86]
Benzyl isothiocyanate	Inhibit mutant p53-p73 interaction	Upregulate and restore function of TAp73, induce LKB1	Breast cancer cells	[87]
LEM2	Inhibit mutant p53-p73 interaction	Restore TAp73 function and prevent MDM2 mediated degradation	Neuroblastoma cancer cells	[88]
Prodigiosin	Inhibit mutant p53-p73 interaction	Restore TAp73 function	Various cancer cell lines	[89]
NSC59984	Inhibit mutant p53-p73 interaction	Restore TAp73 function	Colorectal cancer cells	[90]
PDGFR-β inhibition	Downstream of mutant p53-p73-NF-Y	Prevent PDGFR-β signalling	Pancreatic cancer	[91]
Curcumol	Upregulation p73 + target genes	Enhance TAp73 function	Triple negative breast cancer	[92]
miR-3158	Downstream of mutant p53/p73	Reactivate the miR-3158 signalling pathway	Breast cancer cells	[93]
Pramlintine	Downstream of mutant p53/p63	Inhibit the function of amylin	Lymphomas	[30]
Tamoxifen/inhibition ER β1	Downstream of mutant p53-p63/ER β1	Prevent transcription of GOF associated genes	Triple negative breast cancer cells	[94]
miR-130b	Downstream of mutant p53-p63	Reactivate the miR-130b signalling pathway	Ovarian cancer cells	[95]
Let7i	Downstream of mutant p53-p63	Reactivate the let7i signalling pathway	Breast cancer cells	[96]
Inhibition of TGF β or RAS signaling	Prevent TGF β-induced mutantp53-p63-pSMAD interaction	Reactivate TAp63 function	Various cancer cells	[59]
Inhibition of EGFR, c-MET, ROCK	Downstream of mutant p53-p63	Prevent invasion induced by loss of TAp63 function	Various cancer cells	[47,97]

* Short Interfering Mutant p53 Peptides.

**Table 2 ijms-20-06188-t002:** Compounds that reactivate mutant p53 by restoring protein folding.

Drug	Cells	Reference
Prima-1	Various	[99]
Mira-1	Various	[100,101]
CP-31398	Various	[102,103]
Stima-1	H1299, HCT116, Saos-2 cells	[104]
NSC319726 and metal ion chelators	Various	[105,106]
PEITC *	Various	[107]
Ellipticine	Various	[108]
Phikan083	Glioblastoma	[109,110]
Sch29074	Glioblastoma, various	[110,111]
PK7088	HUH-6, NUGC-3	[112]
Peptide-46	Saos-2	[113]
COTI-2	Breast cancer cells	[114]
Thiol reactive compounds	Various	[115]
Capsaicin	U373, H1299, SK-Br3 cells	[116]
SLMP53-2	Hepatocellular carcinoma	[117]

* phenethyl isothiocyanate.

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
