# Peer review of "The Diverse Functions of Mutant 53, Its Family Members and Isoforms in Cancer"

_ijms, 2019, doi:10.3390/ijms20246188_

Round 1

Reviewer 1 Report

The review by Hall & Muller aims at describing the interplay between different forms of p53 mutants and p63 and p73 proteins. It addresses an interesting point and highlines the numerous questions that remain to be understood. The review is well structured. However, the scientific language is sometimes poor (see below). Moreover, for readers not so well aware of the different forms of p63 and p73 proteins, a figure illustrating these different forms would be useful. Similarly, in the first part of the chapter entitled “The p53 family of proteins”, a summary of how p53 is stabilized upon stress would be useful as a reminder.

 Minor comments

- line 12: “lost p53 function and display”

- line 13: “its” refers to what ?

- line 17: the sentence “…between mutant p53 and family members (of which family ??), including (not the correct word here) other binding partners…” should be rephrased.

- line 23 : “we”: who “we” ??

- line 26: “to regulate the expression of myriad genes, hence working…”

- line 30: the most apparent role of p53 as a tumour suppressor is not (not only) the frequency of its mutation in all cancers, but the high incidence of cancers at early age, in Li-Fraumeni patients.

- line 37: “throughout the whole amino acid sequence of p53”

- lines 39-41 : the sentence “many mutants….” Is not understandable.

- line 49: “characteristics”

- line 51: “each” : “each what ?

- line 65: “a homologues DNA binding domain” should be rephrased

- line 68: “express” should be “induce the transcription”

- line 70: “to regulate the expression of their own set of genes”

- lines 79-80: The two sentences seem to be antinomic.

- lines 85-86: The sentence is not clear, please rephrase.

- line 102: “to regulate the expression of deltaNp63”

- line 107: please remove the dot at the end of the title

- line 110: why the authors chose to discuss about the p53 R175H mutant (please use this notation all over the manuscript) ? What about the other p53 mutants ?

- line 117 : please refer to figure 1 correctly.

- Figure 1b : It would be interesting to localize the R175H mutant on the sequence of mutant p53

- line 138: the first sentence is not a sentence, please rephrase.

- line 149: “pronounced” is not the correct word in this sentence.

- line 150: “quintessentially unfolded 175H mutant” : what does that mean ?

- line 159: “mutant p53” : which one ?

- line 165: “with expression of mutant p53” mRNA expression or protein level, please specify.

- line 168: “than mutant p53 or deltaNp53 alone” is not clear.

- line 171: “how mutant p53 inhibits p63 and p73 function” : which p53 mutant , What about the other p53 mutants ? It is needed here that a comment is added regarding the behavior of the other p53 mutants regarding their interaction with p63 and p73 variants.

- line 188: “complexes”

- line 189: “affinity” of what ?

- line 200: “investigated” should be “shown”

- line 200: “to influence” what: activity, expression, stabilization, binding partners ?

- line 206: “to regulate” what: activity, expression, stabilization, binding partners ?

- line 208-213: the conclusion of the paragraph seems to indicate that all GOF is due to change in the interaction of the mutant with P§” and/or p73, which is probably not true, please add a comment regarding this.

- line 219: please rephrase “mutated through a variety of other regions”

- line 233: “their target gene promoters”

- line 226: how do aggregates potentially cause metastasis, please add a comment regarding this.

- line 233; the sentence is not clear

- line 234: the sentence is not clear

- line 236: mutant p53 does not interact with other proteins “to” do something, there is no willingness, please rephrase.

- line 239: “target genes” of what : p53 or other transcription factors ?

- line 248: please explain what exactly has ne “noted”.

- line 248-250: “Loss of….family members” : the sentence is not clear.

- line 291: “to promote the expression of the p53 target genes”

- line 302: “interaction with NF-Y” : it is a direct protein-protein interaction or a consequence of the activation of a signaling pathway : please specify.

- line 328: “to inhibit their functions” : the function of what ?

Author Response

The review by Hall & Muller aims at describing the interplay between different forms of p53 mutants and p63 and p73 proteins. It addresses an interesting point and highlines the numerous questions that remain to be understood. The review is well structured. However, the scientific language is sometimes poor (see below). Moreover, for readers not so well aware of the different forms of p63 and p73 proteins, a figure illustrating these different forms would be useful. Similarly, in the first part of the chapter entitled “The p53 family of proteins”, a summary of how p53 is stabilized upon stress would be useful as a reminder.

 We thank the reviewer for thoroughly checking the manuscript. We have tried to clarify paragraphs and sentences that were not clear to the reviewer and we implemented the majority of the suggested changes.

 Minor comments

- line 12: “lost p53 function and display” We have changed the sentence to: ‘Mutant p53 proteins lose p53 function, display dominant-negative (DN) activity and display gain-of-function (GOF) to varying degrees’

- line 13: “its” refers to what ? We have changed the sentence to: ‘GOF was originally attributed to mutant p53’s inhibitory function over the p53 family members p63 and p73’

- line 17: the sentence “…between mutant p53 and family members (of which family ??), including (not the correct word here) other binding partners…” should be rephrased. We have changed the preceding sentences and this sentence to: ‘It has become apparent that this is not the only way in which mutant p53 operates as a large number of transcription factors that are not related to p53 are activated upon mutant p53 binding. This raises the question to what extent mutant p53 binding to p63 and p73 plays a role in mutant p53 GOF’

- line 23 : “we”: who “we” ?? This is meant as ‘researchers’ in general.To avoid confusion we have changed this sentence to a passive tense: ‘Forty years on, two other family members, p63 and p73, a large variety of different isoforms with altered N- and C-termini and last, but not least, thousands of different p53 mutant proteins have been identified’.

- line 26: “to regulate the expression of myriad genes, hence working…” This has been changed

- line 30: the most apparent role of p53 as a tumour suppressor is not (not only) the frequency of its mutation in all cancers, but the high incidence of cancers at early age, in Li-Fraumeni patients. We agree with the reviewer and have changed the sentence into ‘The importance of p53 in tumour suppression is apparent when looking at the frequency of p53 mutations in all cancers, the early onset and frequency of cancers in Li-Fraumeni patients and the many mouse models that display tumourigenesis when p53 is lost or mutated’

- line 37: “throughout the whole amino acid sequence of p53”This has been changed

- lines 39-41 : the sentence “many mutants….” Is not understandable. We have changed this sentence and the preceding sentence into the following sentence: ‘The most common p53 mutants (the hotspots) can be regarded as inactive cousins of the wildtype protein, but screens in yeast have indicate that not all mutant p53 proteins are completely impaired for wildtype function. A large variety in the extent to which each mutant p53 protein can still operate as a wildtype molecule exists’

- line 49: “characteristics” We have changed the sentence into the following: ‘Halevy et al, first described the three major differences between mutant p53 and wildtype p53. Mutant p53 proteins lose the function of tumour suppressor, have an increased transforming potential and show increased stability. These functions are independent of each other and are not exhibited by each mutant to the same extent’

- line 51: “each” : “each what ? see above comment

- line 65: “a homologues DNA binding domain” should be rephrased this has been rephrased as similar and are now displayed in figure 1

- line 68: “express” should be “induce the transcription” this has been changed

- line 70: “to regulate the expression of their own set of genes” this has been changed

- lines 79-80: The two sentences seem to be antinomic. We have tried to clarify this by changing the sentences slightly into: ‘This is reflected in the phenotypes of Tp63 KO mice and Tp73 KO mice that show lethality during late gestation or early after birth due to skin and craniofacial epithelial defects [31, 32]or neurological defects, respectively’

- lines 85-86: The sentence is not clear, please rephrase. We have removed these sentences as they fall outside the scope of this review and a proper explanation of the contradicting results of each of the papers referred to, would substantially increase the length of this review

- line 102: “to regulate the expression of deltaNp63” this has been changed

- line 107: please remove the dot at the end of the title this has been changed

- line 110: why the authors chose to discuss about the p53 R175H mutant (please use this notation all over the manuscript) ? What about the other p53 mutants ? Most studies related to p53 folding and interaction with p63/p73 have been performed using the R175H. We discussed this a bit better later on in this section. However, we do agree that at this point it might be a bit distracting and we have therefore rearranged section 3 to start with why we are looking at R175H.

- line 117 : please refer to figure 1 correctly. this has been changed

-Figure 1b : It would be interesting to localize the R175H mutant on the sequence of mutant p53 This has been added

- line 138: the first sentence is not a sentence, please rephrase. Due to the figure, part of this sentence had weirdly reformatted. We apologise for the mistake and adjusted the sentence to ‘and the DBD of the R175H mutant p53 molecule was sufficient to interact with p73aand p73c

- line 149: “pronounced” is not the correct word in this sentence. We have checked the sentence and we are happy with the word ‘pronounced’ in this sentence. The binding of structural mutants to p63 and p73 is more pronounced than the binding of mutants that are normally folded.

- line 150: “quintessentially unfolded 175H mutant” : what does that mean ?  Quintessential means ‘representing the most perfect or typical example of a quality or class’ Given that mutant p53 R175H is the most studied mutant in relation to binding to p63 or p73, we feel that this is the right description for this mutant. We have moved this sentence to the beginning of section 3 to make it clear why we refer to this mutant in particular.

- line 159: “mutant p53” : which one ? We have added ‘to mutant p53 (R172H or R270H) in promoting invasion and metastasis’

- line 165: “with expression of mutant p53” mRNA expression or protein level, please specify. We have changed this into protein expression

- line 168: “than mutant p53 or deltaNp53 alone” is not clear. We have simplified the sentence to ‘As we are only beginning to understand the function of mutation of p53 in the full length variant of the p53 molecule (TAp53a), at the minute it is only guesswork on how intrinsic mutations affect DNp53’

- line 171: “how mutant p53 inhibits p63 and p73 function” : which p53 mutant , What about the other p53 mutants ? It is needed here that a comment is added regarding the behavior of the other p53 mutants regarding their interaction with p63 and p73 variants. ‘These studies were mostly done with the R175H mutant p53, although some other mutants were investigated and did show similar phenotypes’

- line 188: “complexes” this has been changed

- line 189: “affinity” of what ? this sentence has been changed into ‘The topoisomerase TopBP1, for instance, promotes the formation of both complexes by binding p53 and increasing the affinity for mutant p53 to bind to p63 or p73’

- line 200: “investigated” should be “shown” this has been changed

- line 200: “to influence” what: activity, expression, stabilization, binding partners ? In this sentence it is not important how the binding partners regulate p63 or p73, but that they only regulate one of the two. How they regulate p63 or p73 is detailed in the next few sentences. We have re-worded this sentence to make this clear: ‘Most binding partners have been shown to regulate only p63 or p73’

- line 206: “to regulate” what: activity, expression, stabilization, binding partners ? We have changed the sentence into ‘For all these binding partners it will be interesting to see to what extent they regulate the transcriptional function of just one of the p53 family members or whether they can also regulate the other family member’

- line 208-213: the conclusion of the paragraph seems to indicate that all GOF is due to change in the interaction of the mutant with P§” and/or p73, which is probably not true, please add a comment regarding this. We agree with the reviewer and have changed this paragraph to the following: ‘These findings show that the interacting proteins of mutant p53-family member complexes in cancer can easily modulate some aspects of mutant p53 GOF. This regulation, if appropriately targeted, could be used to reduce the GOF of mutant p53 that is related to p63 and p73 function. In particular, looking at proteins which affect both complexes in the same manner could be interesting. Inhibition of TopBP1, or restoration of ANKRD11 expression would both lead to wild type p63 and p73 signaling which would suppress metastasis and induce apoptosis’

- line 219: please rephrase “mutated through a variety of other regions” We have changed this sentence to: ‘Xu et al proposed a region around amino acid 254 to be required for aggregation of p53 in vivo, but in vitro aggregation was shown to depend on multiple regions and could still occur when this region was mutated’

- line 233: “their target gene promoters” this has been changed

- line 226: how do aggregates potentially cause metastasis, please add a comment regarding this. This is explained in the following sentence ‘With the notion that p53 can form aggregates comes the hypothesis that p53 might be a prion protein. In this hypothesis, p53 protein aggregates would grow until causing toxicity and cell death, leading to release of the aggregates in the cell surroundings and spread to other cells’

- line 233; the sentence is not clear, line 234: the sentence is not clear, line 236: mutant p53 does not interact with other proteins “to” do something, there is no willingness, please rephrase, line 239: “target genes” of what : p53 or other transcription factors ? We have changed this paragraph and all the sentences into the following: ‘Most likely, aggregation cannot be fully responsible for all gain-of-function that we see upon mutant p53 expression. By aggregating, the mutant p53 complex with all of its binding partners is rendered inactive. Although mutant p53 inhibits TAp63 in activating canonical p53 target genes, some reports describe that through mutant p53, TAp63 binds to a unique set of promotors to promote gene expression [69, 70]. In addition, mutant p53 interacts with a wide variety of other transcription factors and proteins to promote invasion, acquire chemoresistance or to enhance tumourigenesis [13, 71]. These include, but are not limited to HIF-1, ETS members and SREBPs [72-74]. Most strikingly, an interaction of mutant p53 with many of these transcription factors potentiates the function of these transcription factors, illustrating that mutant p53 or the mutant p53 complex is not always inactivated through aggregation’.

- line 248: please explain what exactly has ne “noted”. This sentence has been changed into ‘These models generally show more metastasis and a broader aggressive tumour burden compared to p53 knockout mice, but differences between the potency of different mutant p53’s have been noted’

- line 248-250: “Loss of….family members” : the sentence is not clear. This has been changed in to ‘Loss of TAp63 and loss of TAp73 expression have been shown to cause a similar increase in tumourigenesis as seen in mutant p53 models. Given that mutant p53 can inhibit these family members as well as enhancing the function of other transcription factors, this raises the question to what extent mutant p53 exerts it’s GOF through inhibition of its family members’

- line 291: “to promote the expression of the p53 target genes” this has been changed

- line 302: “interaction with NF-Y” : it is a direct protein-protein interaction or a consequence of the activation of a signaling pathway : please specify. This has been changed into via a protein-protein interaction with NF-Y’

- line 328: “to inhibit their functions” : the function of what ? This has been changed into ‘p63 and p73 function

Reviewer 2 Report

Review Report

Ijms-654766 review article

The function of Mutant p53 and p53 family members in cancer; Callum Hall and Patricia A.J. Muller

Hall and Muller have composed a comprehensive review on the functions of mutant 53 and other p53 family proteins in cancer.

The content of the article appears to be adequate for the description of the stated phenomena in both the quality and amount of data.

Specific comments

Title: To better reflect the content of the manuscript my suggestion:

Diverse functions of mutant p53 and other p53 family protein isoforms in cancer

Figures and tables: The quality of the figures and tables should be improved.

For the sake of completeness I suggest to add a second table.  

Figure 1: fuzzy appearance, low contrast, bad legibility.

To improve quality: sharpen fuzzy appearance, increase contrast, bold characters to enhance legibility.

Figure 2: fuzzy appearance, low contrast (black letters on dark grey background), bad legibility.

To improve quality, increase contrast, sharpen letters, increase font size.

Table 1: floppy appearance. To improve quality, compress into regular excel table, e.g.

Table 2: To cover all aspects of mutant 53 in this review, add table 2 listing all compounds restoring the folding of mutant p53, as mentioned on line 309.

Minor comments:

Line 56: grammar, remove comma

             …stresses that stabilise the wildtype p53 protein also promote mutant p53 stabilisation…

Line 68: rewording required

             TA isoforms can bind to canonical p53 target sequences and induce p53 target genes.

Line 69: grammar, transcriptionally inactive

Line 71: vague statement, either add reference or remove part of sentence

                               … with more still being identified.

Line 82 to 86: unclear statements. Both sentences appear to suggest redundancy of functions. No exacerbation of lethal phenotype in double mutants would point to redundant function.

Line 138: unfinished sentence. which molecule binds to p73a and p73x?

Line 180: grammar, remove comma

                               Other proteins prevent the binding of mutant p53 to p53 family member.

Line 187: This sentence is continued from line 172, no new paragraph.

                                Some of these binding partners….

Line 282: Rewording required: "When looking in table 1" sounds a bit oafish.

Remarkably, there are quite a few drugs that prevent..., but no therapeutics that directly interfere...

Line 299: unclear statement. p63 levels may have been under the detection limit?

Line 304 to 306: word redundancy: three times "affect" in one sentence

Line 309: To cover all aspects of mutant 53 in this review, add table 2 listing all compounds restoring the folding of mutant p53.

As listed in table 2, several compounds have been described to induce refolding of mutant p53 thereby restoring wild-type function.

Author Response

We thank the reviewer for his/her suggestions. We have implemented all changes, including a title change that is close to the reviewer’s suggestion ‘The diverse functions of mutant 53, family members and isoforms in cancer’ and added table 2

Figure 1: fuzzy appearance, low contrast, bad legibility.To improve quality: sharpen fuzzy appearance, increase contrast, bold characters to enhance legibility.Figure 2: fuzzy appearance, low contrast (black letters on dark grey background), bad legibility. To improve quality, increase contrast, sharpen letters, increase font size.

We apologise the version you received had images of low quality. The version we uploaded and proofed did not appear to have fuzzy images, but we have ensured that the newly uploaded version has a higher resolution.

Table 1: floppy appearance. To improve quality, compress into regular excel table, e.g.

 The table has been changed      

Table 2: To cover all aspects of mutant 53 in this review, add table 2 listing all compounds restoring the folding of mutant p53, as mentioned on line 309.

A table 2 as suggested has been added to the manuscript

Minor comments: 

Line 56: grammar, remove comma

             …stresses that stabilise the wildtype p53 protein also promote mutant p53 stabilisation…

The comma has been removed

Line 68: rewording required

             TA isoforms can bind to canonical p53 target sequences and induce p53 target genes.

This has been changed into ‘TA isoforms can bind to canonical p53 target sequences and can upon overexpression induce the expression of p53 target genes’

Line 69: grammar, transcriptionally inactive

This has been changed

Line 71: vague statement, either add reference or remove part of sentence

                               … with more still being identified.

We have removed this part of the sentence

Line 82 to 86: unclear statements. Both sentences appear to suggest redundancy of functions. No exacerbation of lethal phenotype in double mutants would point to redundant function. As this is outside the scope of this review, we felt it better to remove these sentences. We felt it would take too much space to explain this properly and credit the referred work appropriately.

Line 138: unfinished sentence. which molecule binds to p73a and p73x? We apologise for the mistake. The figure caused the sentence to be split in an awkward manner. The sentence is now as follows: ‘For p73, a p73 molecule containing the DBD + OD was sufficient to bind to mutant p53 175H and the DBD of the R175H mutant p53 molecule was sufficient to interact with p73aand p73c

Line 180: grammar, remove comma

                               Other proteins prevent the binding of mutant p53 to p53 family member.

This has been changed

Line 187: This sentence is continued from line 172, no new paragraph.

                                Some of these binding partners….

This has been corrected

Line 282: Rewording required: "When looking in table 1" sounds a bit oafish. 

Remarkably, there are quite a few drugs that prevent..., but no therapeutics that directly interfere...This suggestion has been used

Line 299: unclear statement. p63 levels may have been under the detection limit? This has been rephrased as proposed

Line 304 to 306: word redundancy: three times "affect" in one sentence This has been changed into: ‘It would therefore be interesting to systematically determine how each of the compounds that interfere in the mutant p53-p73 interaction affect levels and function of all p53 family isoforms, the interaction of mutant p53 with other transcription factors and the extent of GOF of mutant p53 (e.g. chemoresistance, invasion and metabolism)’

Line 309: To cover all aspects of mutant 53 in this review, add table 2 listing all compounds restoring the folding of mutant p53.As listed in table 2, several compounds have been described to induce refolding of mutant p53 thereby restoring wild-type function.

A table 2 with compounds affecting p53 folding and causing reactivation of p53 function has been added to this review